# Defining Signatures of Arm-Wise Copy Number Change and Their Associated Drivers in Kidney Cancers

**DOI:** 10.3390/ijms20225762

**Published:** 2019-11-16

**Authors:** Graeme Benstead-Hume, Sarah K. Wooller, Jessica A Downs, Frances M. G. Pearl

**Affiliations:** 1Bioinformatics Lab, School of Life Sciences, University of Sussex, Falmer, Brighton BN1 9QG, UK; g.benstead@sussex.ac.uk (G.B.-H.); S.K.Wooller@sussex.ac.uk (S.K.W.); 2Division of Cancer Biology, Institute of Cancer Research, Chester Beatty Laboratories, 237 Fulham Road, London SW3 6JB, UK; Jessica.Downs@icr.ac.uk

**Keywords:** aneuploidy, copy number, non-negative matrix factorisation, mutational signature, machine learning

## Abstract

Using pan-cancer data from The Cancer Genome Atlas (TCGA), we investigated how patterns in copy number alterations in cancer cells vary both by tissue type and as a function of genetic alteration. We find that patterns in both chromosomal ploidy and individual arm copy number are dependent on tumour type. We highlight for example, the significant losses in chromosome arm 3p and the gain of ploidy in 5q in kidney clear cell renal cell carcinoma tissue samples. We find that specific gene mutations are associated with genome-wide copy number changes. Using signatures derived from non-negative matrix factorisation (NMF), we also find gene mutations that are associated with particular patterns of ploidy change. Finally, utilising a set of machine learning classifiers, we successfully predicted the presence of mutated genes in a sample using arm-wise copy number patterns as features. This demonstrates that mutations in specific genes are correlated and may lead to specific patterns of ploidy loss and gain across chromosome arms. Using these same classifiers, we highlight which arms are most predictive of commonly mutated genes in kidney renal clear cell carcinoma (KIRC).

## 1. Introduction

One of the most striking features of the cancer cell genome is the frequently observed abnormal karyotype. Many factors can lead to abnormal structural rearrangements of chromosomes including errors in cell division such as spindle assembly checkpoint defects [1] and missegregation due to issues such as telomerase insufficiency [2]. The inaccurate repair of DNA double-strand breaks can also result in translocations, duplications, deletions and inversions of DNA leading to genome instability [3]. Recurrent translocations are frequently observed in haematological malignancies where the resulting fusion genes drive tumourigenesis [4]. Loss of heterozygosity (LOH) can also contribute to the loss of function of tumour suppressor genes [5], and large-scale copy number changes can lead to oncogene amplification [6,7]. 

In addition, defects in the fidelity of chromosome segregation can lead to the gain or loss of entire chromosomes. The chromosoma can vary greatly within tumours [8] and changes in ploidy are known contributors to tumourigenesis and the progression of certain cancer types [9]. Indeed some individual gene variations such as alterations in *BRAF* [10], *TP53* [11,12], *PTEN* [13] and *VHL* [14] have been linked directly with genome stability and changes in the aneuploidy state of the cell.

The production of large-scale cancer sequencing projects, such as the The Cancer Genome Atlas (TCGA) [15], now available via the Genomic Data Commons (GDC) [16] and The International Cancer Genome Consortium (ICGC) [17], has enabled the detailed interrogation and analysis of the cancer genomes. The multi-omic data sets generated have allowed researchers to identify driver genes [18], investigate correlations between expression, copy number variance (CNV) and mutation data [19] and to calculate the mutual exclusivity of altered gene sets [20]. 

CNVs have been used as predictive features in previous studies in an attempt to characterise distinct classes of tumour and provide insight into the functional significance of alterations across the cancer genome. Zack et al. [21] found recurrent copy number aberrations in a number of pan-cancer regions where no oncogene or tumour suppressor had previously been described. They suggested that recurrently deleted regions could either be enriched for novel tumour suppressors or alternatively enriched with non-essential genes. Kim et al. [22] revealed similarity of chromosomal arm-level alterations among developmentally related tumour types as well as a number of co-occurring pairs of arm-level alterations. 

Ciriello et al. [23] have postulated that tumours can be classified as either M class or C class, that is, those driven primarily by mutations (M class) or copy number aberrations, often occurring alongside mutations specifically in *TP53* (C class). In their study C class tumours include breast (BRCA), ovarian (OV), lung (LUSC) and head and neck squamous cell (HNSC) carcinomas. M Class tumours include kidney clear-cell carcinoma (KIRC), glioblastoma multiformae (GBM), acute myeloid leukemia (LAML) and colorectal carcinoma (COADREAD).

There are three subtypes of kidney renal cancers, KIRC, kidney renal papillary cell carcinoma (KIRP) and kidney chromophobe (KICH). KIRC is the most common form of kidney cancer and the 8th most common form of cancer in the UK. KIRC often presents a distinctive, previously described karyotype. KIRC tumours are often initiated by loss of function of *VHL* caused by a combination of mutation or epigenetic silencing of the gene in one allele, in conjunction with the loss of heterozygosity of chromosomal arm 3p, where the gene resides [24]. For complete transformation into a cancer cell, further abnormalities are required. These commonly include mutations in *PBRM1*, *BAP1* or *SETD2* all of which are also located on chromosome arm 3p [25]. Other characteristic arm-wise copy number changes in KIRC tumours include losses in 8p, 9p and 14q and gains in 5q and 7q [26].

In this study, we focused on the analysis of kidney cancers, giving us the opportunity to compare three molecularly distinct cancers all arising at the same primary site. In addition, KIRC has a distinct and well characterised karyotype, aiding the validation of our methods. We utilised publicly available copy number data to compare the overall copy number status of cancer cell lines across tissue types and to identify correlated arm-wise copy number changes. We used unsupervised machine learning to discover recurring patterns of chromosomal arm ploidy change across each cell line and tissue type. Using these insights, we investigated the association of commonly mutated genes in kidney cancers with the general aneuploidy status of samples across our three kidney cancer tissue datasets. Finally, to explore whether specific mutated genes could lead to specific arm-wise copy number variance patterns, we employed a set of supervised machine learning classifiers to measure the predictive power of chromosome arm ploidy profile data to identify mutations in specific genes.

Describing patterns of copy number change across diseases and identifying associated gene mutations may provide clues for the drivers of these genomic instabilities and how specific genes interact with the karyotype as a whole.

## 2. Results

### 2.1. Magnitude of Copy Number Changes Differs Between Cancer Subtypes

We first compared overall chromosomal copy number change across all tissue types. We processed the raw CNV data to calculate mean chromosomal copy number (CCN) for all samples (excluding sex chromosomes) and grouped these samples by cancer subtype (Figure 1).

Across tumours from all tissue types present in this study, we see a slight overall gain of genetic material with a mean CCN of 2.046. There is much variance with a bottom quartile of CCN 1.932 and a top quartile of CCN 2.128. The lowest CCN recorded in our samples is 0.375, from a breast cancer sample, and the highest 8.308 from an ovarian cancer sample.

Mean chromosome copy number did not fall below two for any of the tissue types included in this study. KICH exhibited most gain with a mean CCN of 2.105 (STD = 0.450) as well as the lowest 25th percentile at CCN 1.605 and the highest 75th percentile at CCN 2.488. Ovarian cancer samples showed the most variance of CCN with a standard deviation of 0.499. The tissue types with the lowest variance of CCN included prostate adenocarcinoma and KIRC with standard deviations of 0.175 and 0.219 respectively.

These results show that the extent of genomic ploidy change seems to vary by cancer subtype type rather than by the primary site of the sample. For example, the three classes of kidney tissue exhibited notably different distributions.

### 2.2. Different Genes Are Associated with Changes in Overall Aneuploidy in Different Tissues

We next investigated how gene mutations were associated with overall ploidy change. We identified the 250 most commonly mutated genes in our kidney cancers. Genetic mutations in *VHL*, *PBRM1, SETD2, BAP1, MTOR, KDM5C, PCLO, KMT2C, ARID1A* and *SPEN* were most commonly seen in KIRC and *MET, KMT2C, AR, FAT1, PCLO* and *NF1* mutations were most commonly found in KIRP tissue [27]. Mutational data was not available for KICH tissue.

Within each tissue type, we compared the distributions of copy number change between samples with a specific mutation and those without a mutation to calculate a significance score for each gene. After applying Bonferroni correction to correct for false discovery error, we ranked these genes by the reported corrected significance score. 

In KIRC tissue samples, those with *POLE* mutations were reported to show the most significant loss in ploidy (*p* = 1.53 × 10^−13^) while samples with mutations in *TP53* exhibited the most significant gains (*p* = 1.33 × 10^−12^). Other prominent gene mutations associated with ploidy change in KIRC included *GRM8* (associated with loss, *p* = 3.44 × 10^−12^), *SYNE1* (associated with gain, *p* = 2.36 × 10^−11^) and *ASTN1* (associated with gain, *p* = 2.66 × 10^−11^) all of which are associated with brain function and have not previously been associated with ploidy change or KIRC tissue. Mutations in *VHL* and *SETD2* genes, traditionally associated with KIRC tissue, also result in a significant change in ploidy in our KIRC tissue samples (*p* = 2.00 × 10^−3^ and *p* = 6.00 × 10^−3^ respectively) though mutations in *PBRM1* did not. 

In KIRP tissue, we found that patients with *KRAS* mutations showed the most significant loss in copy number (*p* = 2.2 × 10^−16^) followed by *THSD7B* (*p* = 2.48 × 10^−36^) and *CHD4* (*p* = 3.31 × 10^−24^). Patients with mutations in *EP400* (*p* = 8.99 × 10^−16^) and *PCDH11X* (*p* = 2.79 × 10^−9^) exhibited the most significant levels of ploidy gain. Surprisingly, we found that mutations in *TP53* did not appear to result in any significant ploidy change in the available KIRP tissue samples. 

### 2.3. Chromosome Arm Ploidy Patterns Vary by Tissue

We next investigated copy number data stratified by chromosome arm to ascertain whether certain chromosome arms were preferentially gained or lost in the kidney cell lines compared to other cancers (Figure 2). All *p*-values below were calculated using segment mean values to allow direct comparison. 

Across our kidney tissue samples, we found that each tissue type exhibited a distinct pattern of copy number variance. This suggests that arm-wise copy number change profiles depend on tissue type more than on the primary cancer site.

As expected, based on previous studies 5q showed the highest ploidy gain on average in our KIRC tissue samples (CCN = 2.27, STD = 0.24) and 3p the most loss by a significant margin (CCN = 1.61, STD = 0.23). In comparison, in our pan-cancer data neither 5q (CCN = 1.98, STD = 0.26, *p* = 3.50 × 10^−144^) or 3p (CCN = 1.94, STD = 0.26, *p* = 2.21 × 10^−144^) show any notable copy number variance on average.

In KIRP tissue 7p and 7q both exhibit a significant gain (both CCN = 2.48, STD = 0.46) whilst in our pan-tissue data the arms show less dramatic change (7q CCN = 2.24, STD = 0.35, 7p CCN = 2.23, STD = 0.39). 

KIRP also exhibited loss in 22q (CCN = 1.84, STD = 0.22) while KICH tissue showed a gain (CCN = 2.33, STD = 0.317, *p* = 1.44 × 10^−20^) compared to the pan cancer (CCN = 1.92, STD = 0.29, *p* = 1.71 × 10^−38^).

The KICH arm-wise copy number data exhibit much larger copy number changes across every arm with only 21p appearing relatively diploid. On average each of the other arms seems to have either gained or lost roughly half of its genetic material. These results support existing research into the karyotype of KIRC tissue, including loss in 3p and gain in 5q.

Across the pan-cancer samples, a set of 10 cancer types excluding the three kidney cancers (see methods), the chromosomes that exhibited the most notable copy number change included 8q and 17p. We note that the patterns found in our pan-cancer samples were not shared with our kidney tissue samples.

Chromosome arm 8q was shown to exhibit the highest average copy number gain and variance in our pan-cancer data (CCN = 2.29, STD = 0.49). Comparatively, KIRC and KIRP tissues both showed significantly less change in arm 8q with a mean copy number close to normal (KIRC CCN = 2.04, STD = 0.22, *p* = 1.65 × 10^−87^ and KIRP CCN = 2.00, STD = 0.25, *p* = 2.12 × 10^−55^).

Across our pan-cancer samples chromosome arm 17p showed the greatest loss of ploidy on average (CCN = 1.88, STD = 0.30). Again, our KIRC tissue samples were nearly diploid (KIRC CCN = 2.01, STD = 0.16, *p* = 0.17 × 10^−74^) and KIRP showed a significant gain in copy number (KIRP CCN = 2.27, STD = 0.37, *p* = 7.79 × 10^−55^).

The apparent stability of 8q in KIRC and KIRP tissues compared to our pan-cancer set, may be a possible avenue for further research.

### 2.4. Patterns of Arm Ploidy Correlation Are Found Within Tissue Types

A loss in chromosome arm 3p paired with a gain in chromosome arm 5q is a common and well described trait of KIRC tissue. Next, we investigated how changes of ploidy in other pairs of chromosome arms relate to each other. We measured the correlation coefficient for each pair of arms within tissue type (Figure 3). Significance scores were measured using permutation tests as discussed in methods.

The most prominent positive correlation scores occurred between arms of the same chromosome (For example chromosome arms 1p and 1q commonly exhibited a similar amount of gain or loss together), however these scores were not uniform across all tissue types. In KIRC, for example, the 9p–9q arm pair was reported to show the highest positive correlation coefficient (r = ~0.73, *p* = 4.08 × 10^−72^) while in KIRP it was the 20p–20q arm pair (r = ~0.78, *p* = 7.27 × 10^−53^). Conversely the 9p–9q arm pair in KIRP exhibits a much lower correlation score (r = ~0.31, *p* = 2.69 × 10^−18^) and the 20p–20q arm pair in KIRC shows a slightly negative correlation (r = ~−0.02, *p* = 4.17 × 10^−55^).

In terms of negative correlation, where one arm’s gain of genetic material was commonly associated with a loss in the others, we found the 3p–5q pair featured prominently for KIRP tissue (r = ~−0.63, *p* = 1.37 × 10^−7^) but perhaps surprisingly not as prominent as for KIRC (r = ~−0.13, *p* = 2.21 × 10^−5^) although both are reported to be significant. In KIRC tissue the 4q–6q exhibits the strongest negative correlation ploidy (r = ~−0.58, *p* = 3.64 × 10^−5^) (Appendix A).

### 2.5. Arm Ploidy Signatures Are Somewhat Enriched for Tissue Types 

The patterns of arm ploidy correlation found above suggest that the underlying mechanisms that occur in specific tissues may give rise to different profiles of loss and gain of ploidy in different chromosome arms. In an attempt to capture these profiles more fully, we employed non-negative matrix factorisation (NMF) to generate a number of arm ploidy signatures across our kidney tissue data.

NMF is a multivariate analysis tool commonly used for easily interpretable decomposition. NMF was chosen because it provides both dimensionality reduction and clustering. Essentially NMF decomposes a feature matrix into two descriptive matrices, a basis, which describes the feature composition of each component and a coefficient, which describes the component composition of each sample in the original feature matrix.

NMF generates a specified number of components (used as signatures). To identify the optimum number of components for each dataset, we ran a number of trials for decompositions with an increasing value of required components. Each of these trials provided a cophenetic score, a measure of cluster stability, i.e., how well the clusters obtained by NMF preserved the pairwise distances between the original data points. For each dataset, we selected the lowest component count associated with a local minimum cophenetic score.

Our kidney tissue ploidy data was decomposed into six component signatures (Figure 4). We found that some of these signatures where highly enriched for just one of the three tissue types within the kidney group, KIRC, KIRP and KICH whilst others were more mixed. KIRC samples were most prominent in signature 1 and 4, KIRP samples were dominant in signatures 2, 3 and 6, and KICH was found most prominently in signature 5.

In terms of composition, we found that the feature composition of signature 2, associated with KIRP tissue, exhibited the most notable deficiency of 3p paired with gains in 5q which reflected our correlation analysis. This was paired with large gains in 6p, 15q and 16p.

We found that signature 1, predominantly associated with KIRC, featured gains in a number of arms most notably in 10p, 18q and 19p. Signature 5, enriched for KICH samples, featured gains in a number of chromosome arms that are not present in the other signatures and notably exhibited losses in arms 6p, 6q and 7p. Signature 6, the other signature associated with KIRP tissue, exhibited gains in 3p and 3q paired with some loss in 5p and 5q. 19q and 20p also showed large gains.

### 2.6. The Amount of Signature Composition Change Varies by Gene Mutation

We next used cosine similarity, the cosine of the angle between two signatures represented as vectors projected in a multi-dimensional space, to measure the effect of specific mutations on overall signature composition change.

By taking the median signature composition of patients with and without specific gene mutations and measuring the cosine similarities of these compositions, we were able to rank genes by the distance of signature composition between these groups.

In our kidney cancer data, we found a number of gene mutations associated with an increased distance in signature composition. The genes with the highest cosine similarity (all with cosine similarity scores of ~0.678) in our ranked list are included in Table 1. In our pan-cancer data, we found a different set of genes with some similar functionality listed in Table 1. We found overrepresentation of gene associated with the innate immune system as well as *KDM6B*, *INCENP* and *H2AFJ* which are all related to the chromosome organisation.

### 2.7. Some Gene Mutations Are Enriched in Our Kidney Ploidy Signatures

To find gene mutation enrichment in our six kidney cancer signatures, we first categorised samples into signature groups and counted the frequency of mutations occurring in each patients’ group. We then used permutation tests to measure if the frequency of mutation was significantly higher than would be expected in a random sample of patients. Below, we highlight some of the notable gene mutations that were reported to be significantly enriched in each signature.

Signature 1, the largest signature group, and associated with KIRC tissue, exhibited significant enrichment for *MUC4* (*p* = 8.80 × 10^−03^). *MUC4* is associated with changes in *ErbB2* expression, apoptosis, proliferation, differentiation, and some cancers. In signature 2, which is enriched in KIRP tissue and composed of notable ploidy change in 3p, 5q, 6p, 15q and 16p, *CENPF* (Centromere protein F) was mutated in 4.2% of patients (*p* = 9.30 × 10^−05^) and *DSPP* (Dentin Sialophosphoprotein), a gene associated with calcium ion binding and extracellular matrix structural constituent, was also mutated in 4.2% (*p* = 3.55 × 10^−05^). Signature 4, a group again associated with KIRC and changes in 3p, 3q, 17q and 18p, saw enrichment for *KMT2D* (also *MLL2*), a histone methyltransferase, which was mutated in 6.2% of patients (*p* = 1.68 × 10^−08^). Signature 6, associated with KIRP and changes in 6q, 7p, 19q and 20p, was enriched for *MUC4* (Mucin 4) which occurred in 7.1% of patients (*p* = 4.89 × 10^−06^). Signature 3 did not exhibit any significant enrichment for mutations.

### 2.8. Patterns of Ploidy of Chromosome Arms Are Associated with Specific Mutated Genes

To investigate to what extent mutations in specific genes lead to specific patterns of arm-wise copy number change, we trialled a set of four machine learning classifiers designed to predict gene mutations based on a patient’s chromosome arm ploidy profile (see methods). Using an AUC ROC score of 0.70 as a cut-off, we found that the patterns of ploidy in chromosome arms had varied predictive power for mutations in specific genes. 

We trialled four classifiers; Bernoulli naive Bayes, support vector machine, logistic regression and random forest. Due to consistently better performance when compared to the other classifiers all ROC AUC scores below are based on the results of the random forest classifier. In our pan-cancer dataset patterns of ploidy changes in chromosome arms were strongly predictive of mutations in *VHL* (ROC 0.91), *TP53* (ROC 0.74), and *PBRM1* (ROC 0.71), with *BAP1* (ROC 0.67) narrowly missing the cut-off score. 

To better understand how different features contributed to the predictive power of each classifier, we ranked the importance of each feature for each model. Feature importance was calculated by measuring the mean decrease in classifier accuracy when systematically holding out each variable across all tree permutations in a random forest. Feature importance was reported as a mean decrease in accuracy.

When analysing the important features in the predictive model, losses in 3p featured heavily in patients with *BAP1*, *PBRM1* and *VHL* mutations. Gains in 5p were an important feature for both *PBRM1* and *VHL* mutated genes.

In general, KIRC tissue arm-wise CCN data, which exhibited less variance than KIRP tissue, were less predictive of specific gene mutations. The highest scoring genes, in terms of ROC AUC score, were *PTEN* (ROC 0.68), *PBRM1* (ROC 0.61) and *AKAP9* (ROC 0.61) (Figure 5). 18q previously noted for its overall low ploidy change in KIRC tissue was found to be the most important feature for both *AKAP9* (0.15 Mean decrease in accuracy) and *PTEN* (0.081 Mean decrease in accuracy) (Appendix A).

Chromosome arm 3p, on which both *PBRM1* and *BAP1* are located, commonly exhibits loss in KIRC tissues. However, we found that 3p reported a relatively low overall importance score in our KIRC tissue predictive models. Rather than 3p the highest ranking feature for *PBRM1* was 10p (0.063 Mean decrease in accuracy). This may be due to the all but uniform loss of these genes along with 3p in KIRC tissue (94% of patients with overall loss in 3p and 71% with less than CCN 1.8) leading to less variation and, as such, less signal.

In models derived from our pan-cancer data, features based on patterns of arm-wise CCN performed relatively well when predicting *TP53* and *PTEN* mutations with AUC ROC scores of 0.74 and 0.73 respectively. *TP53* has been implicated in genetic instability in many tissue types [28,29]. *BAP1* also performed fairly well with a ROC AUC score of 0.66; with important features being a loss of 3p (0.076 Mean decrease in accuracy) and of 3q (0.15 Mean decrease in accuracy). Our pan-cancer tissues models performed poorly when predicting *PBRM1* with a ROC AUC score of 0.47. 

Once again both *PBRM1* and *VHL* were found to be associated with KIRC tissue with losses in 3p and gains in 5p featuring as important predictors for both *PBRM1* and *VHL* mutated genes. Chromosome arm ploidy proved a strong predictor of mutations in *VHL*, *TP53* and *PBRM1* in our pan-cancer dataset and the insight drawn from our feature importance scores again matched our expected results.

## 3. Discussion

The goal of this study was to investigate how genome-wide and chromosome arm-wise ploidy varies by tumour type, how these ploidy patterns are associated with genetic mutations and how suitable ploidy data is as a predictor of specific mutations.

As described above, we observed significant variation in both genome-wide and chromosome arm-wise ploidy between samples from different tissue types. This can be expected given that different tumours are driven via gains or losses of specific genes located on various chromosome arms. While we observed a small number of generalisations regarding the pattern of genetic material, e.g., 20p shows gains in CCN across all tumour types, beyond these similarities, there is significant variation in arm ploidy profiles between tumour types.

Our focus on KIRC and kidney tissue was due to its distinct and well-described karyotype. Throughout this study, we compared our observation with previous experimental studies of KIRC to cross-validate our analysis. Turajlic et al. found that KIRC is generally characterised by recurrent copy-number variants in arms including, but not exclusive to, 3p, 5q, 7q, 8p, 9p, and 14q. *SQSTM1*, a gene which resides on 5q, an arm which shows relatively large gains in CCN in this study has been postulated to represent an alternative mechanism for activation of mTORC1 [30]. *PBRM1*, *VHL*, *BAP1* and *SETD2* all reside on 3p, a copy of which is known to be often lost at the outset of KIRC tumourigenesis. As such, we expected changes in 3p ploidy to feature prominently in our results. The association between 3p and KIRC was supported by the findings in this study, both whilst investigating the chromosome arm ploidy profile of KIRC compared to other tissues and while calculating feature importance as part of our classification, where we clearly found 3p performing as an important indicator for both *PBRM1* and *VHL* mutations. We also found some association between 3p and 5q in KIRP tissue samples both in our correlation and signature analysis.

Of further interest is the prominence of 5q in our results which often sees a gain in ploidy. This increased importance may be the result of a number of factors. This gain in ploidy at 5q might be a direct or indirect result of the loss of heterozygosity and resulting vulnerability to loss of gene function at 3p [26]. Our random forest importance scores suggest that 5p is highly associated with both *PBRM1* (0.088 Mean decrease in accuracy) and *VHL* (0.13) mutations across all tissues. Chen et al. [31] describe the gain of 5q as a common trait of papillary-enriched KIRC subtype which often includes mutation or amplification of *MET*. 

*VHL*, a gene often found mutated early during the progression of KIRC and found associated with changes in ploidy in this study, has been shown to drive aneuploidy [32]. Loss of a functioning *VHL* gene (located on chromosome arm 3p) has previously been associated with spindle misorientation, chromosome instability and aneuploidy [14]. Similarly, *PBRM1*, a gene that encodes BAF180 and also located on chromosome 3p, has been shown to be important for the establishment or maintenance of cohesin on chromatin at centromeres. Loss of functioning *PBRM1* has recently been reported as a driver of chromosomal instability and aneuploidy [33].

The observations in this study suggest that our analysis has been sensitive to known karyotypic patterns and may be useful for the detection of additional patterns.

As well as these commonly cited kidney tissue gene mutations, we identified a number of other mutations significantly associated to changes in kidney tissue karyotypes. We identified a number of genes associated with increased overall ploidy change, such as *POLE* in KIRC tissue and *KRAS* in KIRP. We also found several gene mutations associated with the innate immune system and chromosome organisation such as *KDM6B*, *INCENP* and *H2AFJ* that appear to contribute to broad changes in patterns ploidy patterns. Further, we evidenced genes significantly associated with the ploidy pattern signatures that we generated in this study, which in some cases described karotypes shared between tissue types. 

## 4. Materials and Methods

### 4.1. Data Acquisition

In total, we analysed 3,559,315 samples (356,069 in our kidney group, 3,203,246 in our pan-cancer group) across 5756 patients (888 kidney, 4868 pan-cancer). Our source data consisted of somatic mutation data (BCM Curated or Automatic Somatic Mutation Calling) and copy number variant data (BI Genome-Wide SNP6) downloaded from the GDC data portal [16]. 

For our kidney cancer data set, we included samples from KIRC (Renal Clear Cell Carcinoma), KIRP (Kidney renal papillary cell carcinoma) and KICH (Kidney Chromophobe). For our pan-cancer dataset, we included patients from cohorts with tissue types including; BRCA (Breast cancer), DLBC (Diffuse large B-cell lymphoma), GBM (Glioblastoma), LGG (Low Grade Glioma), LUSC (Lung Squamous Cell Carcinoma), MESO (Mesothelioma), OV (Ovarian), PRAD (Prostate Adenocarcinoma), SARC (Sarcoma), UCEC (Uterine Corpus Endometrial Carcinoma).

To compare the aneuploidy exhibited by different tumour types, we used the segment mean values in the CNV data provided by TCGA/GDC. The segment mean value is the log base 2 geometric mean of the ratio a sample’s copy number over the wild-type copy number. Throughout this study, our CCN values were calculated directly from these segment mean values using the formula.

(2^Segment Mean) *2

Sex chromosomes were excluded from the study, thus the chromosomal copy number for healthy tissue is 2. 

To ensure the data represented large-scale changes in ploidy rather than smaller, linear duplication CNVs, we sorted the sample data by length and removed the shortest samples observed in the bottom 25th percentile of data. Samples with a probe number of less than 10 were also filtered out to provide high confidence in the CNV data as per Laddha et al. [34].

All data preprocessing and analysis was completed using R 3.4.4 [35] and the machine learning component of the study was completed using python pandas version 0.19 [36].

### 4.2. Describing Chromosome Arm-Wise Ploidy Patterns

To prepare the data for modelling, we segmented CNV data for each patient by chromosome arm. Centromere position data were obtained from genome.ucsc.edu [37]. These data were organised with each patient as an observation and each of their chromosome arm’s average CCN as a variable. 

### 4.3. Correlation of mutated genes with overall ploidy change

We identified the ten most recurrently mutated genes in KIRC and KIRP tumours via cBioPortal [27]. In KIRC tissue these genes included *VHL* (occurring in 47.8% of KIRC patients), *PBRM1* (34.8%), *SETD2* (12.1%), *BAP1* (9.4%), *MTOR* (6.7%), *KDM5C* (5.9%), *ARID1A* (4.1%), *KMT2C* (4.1%), *SPEN* (3.8) and *PTEN* (3.8). In KIRP these genes include *MET* (7.4%), *KMT2C* (6.4%), *SETD2* (5.7%), *KMT2D* (5.0%), *BAP1* (5.0%), *AR* (4.6%), *FAT1* (4.3%), *PCLO* (4.3%), *PBRM1* (3.9%), *NF2* (3.5%). Little mutation data was available for KICH at the time of writing.

Samples with deleterious mutations that were likely to change the protein product or the functioning of the protein such as missense mutations, frame-shift insertion and deletions, nonsense mutations and in-frame insertions and deletions were labelled as mutations in the data whilst all other samples were labelled as not-mutated.

To calculate correlation between each mutation and overall genome ploidy, we took an absolute, positive value for each sample’s segment mean to better measure and compare overall average ploidy change. This data was grouped by individual patients resulting in a list of all patients each with an average absolute genome-wide segment mean.

A list of patients featuring mutations in our chosen genes was then extracted from the somatic mutation data and used to filter our preprocessed CNV data to calculate the probability of each chosen gene mutation being associated with a general change in ploidy. For each gene, the cohort of patients with a mutation in that particular gene was sampled against a control group of all other available patients.

We used t-tests to calculate the probability that patients with mutations in each gene would exhibit a significant change in overall ploidy. This was repeated in our pan-cancer, KIRC and KIRP tissue datasets for comparison. All *p*-values were adjusted through the use of the Bonferroni procedure to correct for false discovery.

### 4.4. Arm Ploidy Correlations

To measure correlation between chromosome arms, we first stratified patients by tissue type and then measured correlational coefficients between each arm pair for all the samples within those groups. 

Significance values were calculated using a permutation test where a mock distribution was produced using the same data for each arm pair but with one arm’s data randomly permeated for 1000 samples. The real value was then compared to this mock distribution using a student t-test. 

### 4.5. Generating Arm-Wise Ploidy Signatures

Arm-wise ploidy signatures were generated from the arm-wise ploidy data (as above) using non-negative matrix factorisation (NMF). We used a cophenetic correlation coefficient score via the R NMF library [38] to measure the stability of our models and to select the most stable component count. When associating a sample with a signature, we chose the highest scoring component in the coefficient matrix for that sample. 

### 4.6. Gene Mutation and Pattern Change

To measure the distance between the arm-ploidy pattern of groups of patients with and without gene mutations, we took the median values of each arm for each group and measured distance using cosine similarity. We took the top 250 most frequently mutated genes in kidney cancers to analyse. Similar to the arm ploidy correlation analysis to measure significance for each of these gene mutations, we used permutation tests this time randomising the labelling of patients in the two group.

### 4.7. Finding Gene Mutations Enriched Within Signatures

To find the most frequently mutated genes in each signature, we stratified our samples by the most prominent signature (as above), counted the frequency of gene mutations in these groups and ranked by their frequency. To determine whether the frequency of mutations was significantly increased compared to that expected in the case of random association, we once again used permutation tests using randomised sampling of all patients from all signatures to create our mock distribution. In each case the number of randomly sampled patients was equal to the number of samples found in the respective signature.

### 4.8. Using Chromosome Arm Ploidy Patterns to Predict Gene Mutations

A boolean, stating whether the patient suffered a mutation in the respective gene or not, served as the label for each observation. Data was split randomly into training and testing groups with a test size of 0.2 and a training set size of 0.8 of all samples.

To measure the predictive power of chromosome arm-wise segment mean for specific gene mutations, receiver operating characteristic area under the curve (ROC AUC) scores were calculated. We initially trialled four different machine learning classifiers; Bernoulli Naive Bayes, Support Vector Machine, logistic regression and random forest. Hyper-parameters for each classifier were optimised using 5 fold cross validation.

Ultimately it was found that a random forest classifier with 1000 estimators and a minimum sample leaf size of 30 performed consistently better when compared to the other classifiers and so ROC AUC scores given in this study were all a result of this classifier

For each model, feature importance was calculated by measuring the mean decrease in classifier accuracy with the removal of each feature across all trees in the random forest. This metric was reported as the mean decrease in accuracy given the removal of a feature.

Using this measure of the mean decrease in accuracy, we ranked and identified chromosome arms that were most commonly lost or gained when specific genes are mutated. This analysis was applied to KIRC tissue data alone and finally to all tissue types excluding KIRC for comparison (Appendix A).

The source code and data for this study is available via bitbucket at https://bitbucket.org/bioinformatics_lab_sussex/ploidy_nmf. The source data required for running this analysis, including copy number and mutation data is available at: https://bitbucket.org/bioinformatics_lab_sussex/ploidy_nmf/downloads/.

## Figures and Tables

**Figure 1 ijms-20-05762-f001:**
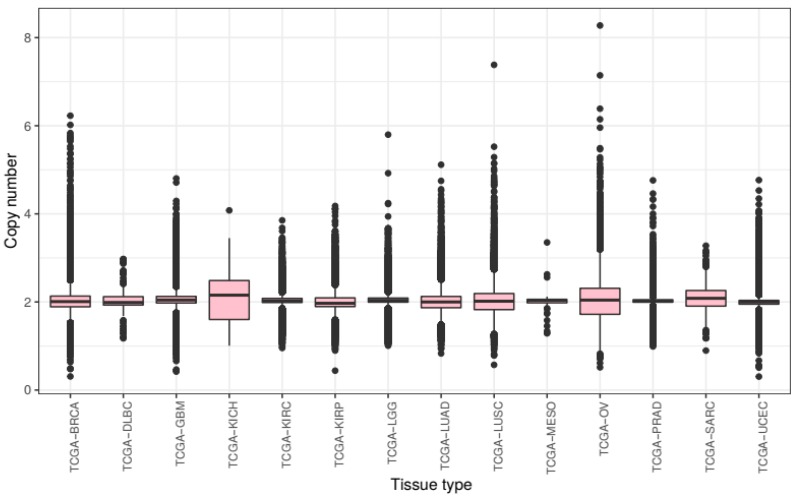
Chromosomal copy number by tissue type—A box-plot summarising mean genome-wide copy number for all samples in the 14 tissue types featured in our pan-cancer dataset. Copy number values were converted from segment mean sourced via TCGA/GDC data. A legend for the featured tissue types is available in the methods section.

**Figure 2 ijms-20-05762-f002:**
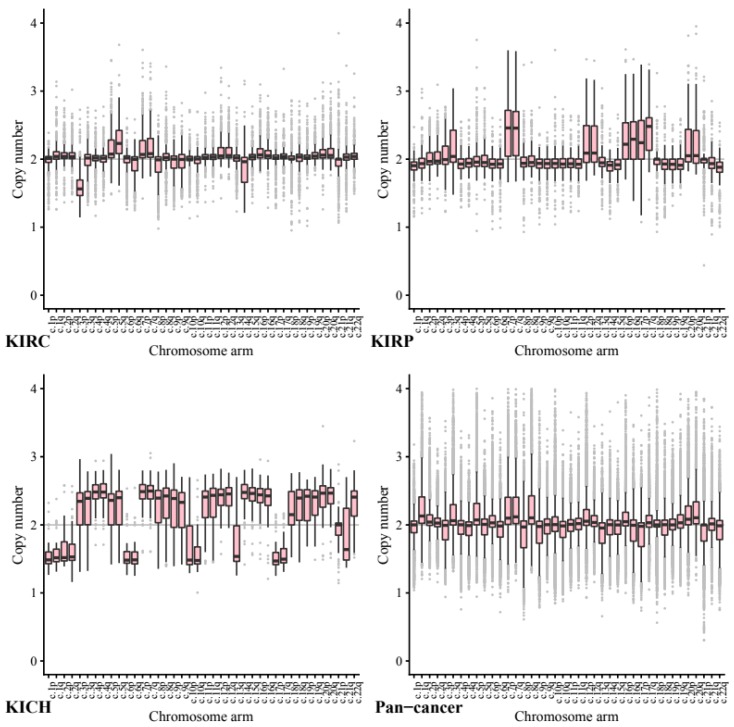
Chromosome Arm-wise copy number. By grouping chromosome arm-wise copy number data across our kidney tissues individually, we develop a clearer picture of the pattern of chromosome arm copy number that occurs specifically in each kidney tissue compared to our pan-cancer dataset.

**Figure 3 ijms-20-05762-f003:**
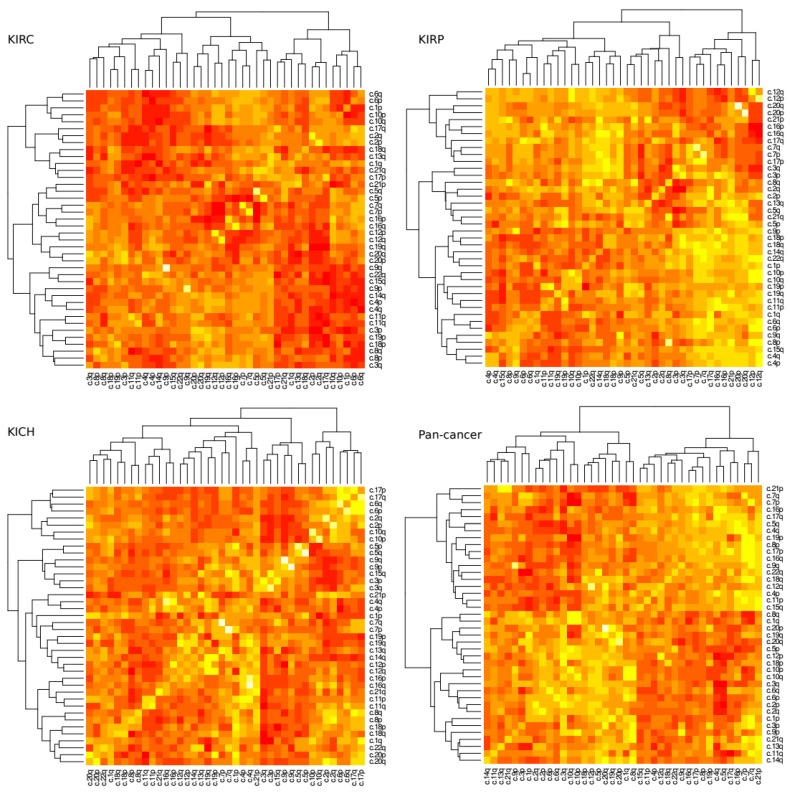
Arm ploidy correlational heat maps. Taking the Pearson correlation coefficient as our measure, we visualise the correlation of gain and loss of copy number for each chromosome arm pair in each of our kidney datasets and in our pan-cancer data. Bright yellow tiles denote highly positive correlation and dark red tiles denote highly negative correlations. The results along the diagonal (which report, for example, of the correlation of c.1p–c.1p) are normalised to *r* = 0 to improve overall contrast).

**Figure 4 ijms-20-05762-f004:**
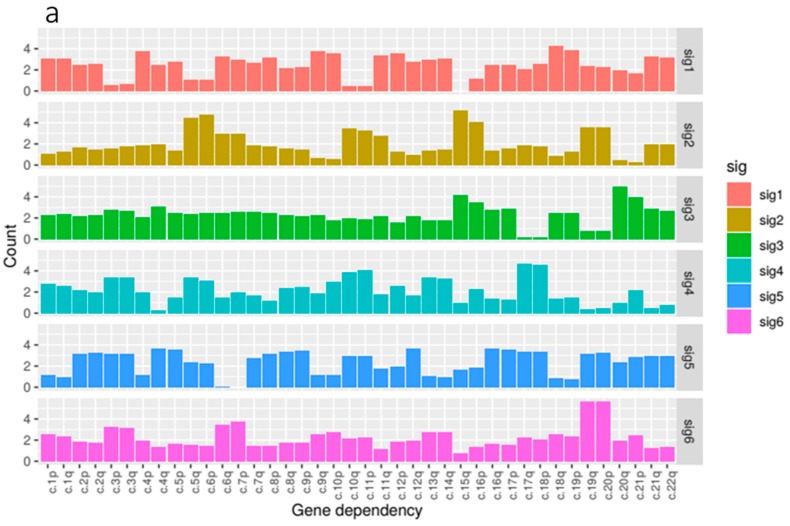
(**a**) Copy number signature composition. A breakdown of the features that describe each of our six signatures. For example, we can see that chromosome arms 19q and 20p are prominent features in signature 6. (**b**) A chart showing the proportion of kidney patients categorised by most prominent signature and stratified by tissue type. From this visualisation, we can see that signature 1 is highly enriched for KIRP tissue and signature 5 is highly enriched for KICH tissue.

**Figure 5 ijms-20-05762-f005:**
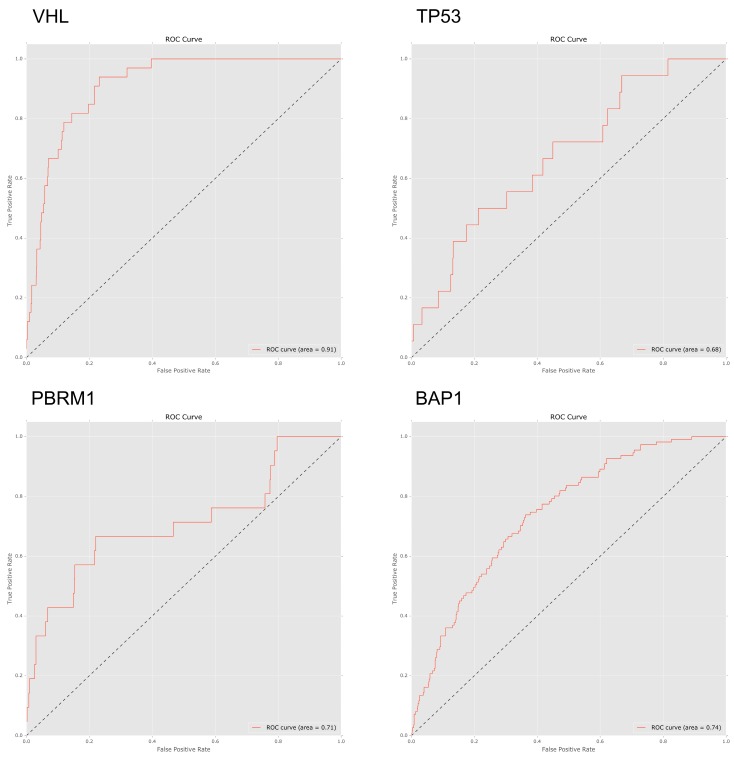
AUC ROC curves used to measure the performance of random forest classifiers trained on arm-wise chromosome copy number patterns to predict gene mutation status of *VHL*, *TP53*, *PBRM1* and *BAP1*. A larger area under the ROC indicates better performance.

**Table 1 ijms-20-05762-t001:** Gene mutations associated with an increased distance in signature composition between wild type and mutated kidney cancer samples/mutated pan-cancer samples.

*Gene*	Functional Associations
**Mutated Kidney Cancer Samples**	
*PON2*	Oxidative stress protection Pathogenic bacteria protection
*IL6ST*	Adipogenesis Innate immune system
*KDM6B*	Chromatin organisation control Gene silencing
*SEC16B*	Organisation of transitional endoplasmic reticulum sites and protein export
*INCENP* (Inner Centromere Protein)	A component of the chromosomal passenger complex (CPC), a key regulator of mitosis
**Mutated Pan-Cancer Samples**	
*RNF185* (Ring Finger Protein 185)	Ligase activity selective mitochondrial autophagy
*TMEM30C*	Innate immune system
*CUEDC2*	DNA damage response
*H2AFJ*	Histone H2A.J
*PRG3*	Innate immune system

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
