# Peer review of "Defining Signatures of Arm-Wise Copy Number Change and Their Associated Drivers in Kidney Cancers"

_ijms, 2019, doi:10.3390/ijms20225762_

Round 1

Reviewer 1 Report

The authors of "Defining signatures of arm-wise copy number change and their associated drivers in kidney cancers" aim to investigate the variation of arm-wise ploidy, its patterns as associated to genetic mutations and its value as predictor of specific mutations.

This is an interesting work with some merit and a lot of potential in a field relatively unexploited for disease and mutations classification. Unfortunately the manuscript is very poorly drafted and, even if I do not find major flaws in overall methods, it needs a major, careful revision in the narrative, structure, figures and supplementary materials reference, legends and their embedding into main text. The manuscript in its present form resembles more a draft than a manuscript ready for submission.

Since the topic is of interest for a general biomedical public readability needs to be greatly improved making clearer in the rationale the advantages of using specific methods, explaining what they do and what their purpose is. I try to be more precise with the following two examples.

Example 1:
Line 275: "When analysing important features in the predictive models", the reader does not know what "important" means. Is it a general "importance" or is something specific? Moreover, HOW "important" features are analysed? A few lines after the reader may understand that the "Gini importance" is involved but this is abruptly introduced and never previously explained in the text. This is relevant since in the discussion there is ample reference to this concept, that should be made clearer to the reader as the ML methods used to highlight predictive power (as stated also in the abstract). Thus, do not limit explanation of the value of "Gini importance" to the methods section but also introduce it in the results when needed.

Example 2:
In-text lengthy lists of genes in paragraphs 2.6 and 2.7 are confusing and not really useful to support the points which are the link between signature composition and gene mutations, and their enrichment. In my opinion these two paragraphs could be merged into one and gene lists with associated values could be put into tables, leaving commentary and description in the text.

Example 3:
The cophenetic correlation obtained on NMF components (lines 198-203) is the metric used to select component signatures: once again it's not explained how these method is able to summarize and group arm ploidy signatures. A simpler yet complete explanation is needed, stating the goal (summarizing the ensemble of arm ploidy signatures into a manageable number to see if kidney cancers are characterized by specific patterns) and the rationale used by the chosen method to do that (NMF, definition of cophenetic score, cluster stability, ...).

Other important inconsistencies are:
1) All figures need to be more readable in axes (too small as they are generated)
2) Figure 3, The correlation clusters in these heatmaps are difficult to grasp at first sight, due to small characters and lack of the color scale legend
3) Supplementary figures 1 and 2 COMPLETELY lack legend. What are these?
4) ALL S1 tables, as referred to in lines 281, 450, 451, are completely missing. No traces of them either in the main text, or in the supplementary zip archive (or even in the bitbucket repository).
5) Supplementary material in the bitbucket repo is not described at all, just linked-referenced. All supplementary material - if provided - must have a clear description and be linked to main text where is appropriate.

Additional points:
6) In the general narrative it's not clear enough why kidney cancers were chosen as the investigation objects: is there any contradictory data? are these cancers particularly prone to this kind of investigation, and why? In the discussion I found (line 313) an explanation (i.e. distinct and well described karyotype), thus this should immediately made clear in the introduction to justify kidney cancer selection as a manageable proxy for such study.
7) It's not completely clear to me the relevance internal reletionship between of ploidy changes in pairs of chromosomes (par. 2.4). This feature need to be contextualized and biological relevance let emerge (allelic effects? dominant-recessive effects?).
Moreover the figure does not help: I guess the two axes (please label them) of each heatmaps refer to two pairs. So this is why the value in the diagonal (bears different colors), differently to what happens when the focus is to the comparisons among arms of different chromosomes. Am I interpreting correctly?
8) Figure 4: the two panels (left and right, or a and b) are inverted. Panels should be labelled and put in order coherent with the legend. Moreover the colors used in the two panels are probably from the default ggplot2 palette, but in this case they are internally inconsistent, making the figure a bit ambiguous: in left panel the three colors refer to the kidney cancers, in right panels the same colors do not.

Minor points/typo by line:
- line 47: a leftover underscore "_"
- lines 155-156 are redundant as a repetition of statements in lines 139-141
- in line 169 "post-cancer", should read "pan-cancer", otherwise the sentence/claim does not make sense to me.
- line 205, a leftover "was" at the end of the line
- line 217-218: these description could be much clearer if the described object would be pointed in the figure by - for instance - a set of markers or a set of arrows.
- in figure 5 are shown ROCs for performance of random forest: what about the other classifiers cited? Are they consistent?
- line 343: typo "kideny" in place of "kidney"
- line 425: typo "them" in place of "their"

Author Response

Response for Reviewer 1

We have highlighted our responses in yellow

Since the topic is of interest for a general biomedical public readability needs to be greatly improved making clearer in the rationale the advantages of using specific methods, explaining what they do and what their purpose is. I try to be more precise with the following two examples.

Example 1:
Line 275: "When analysing important features in the predictive models", the reader does not know what "important" means. Is it a general "importance" or is something specific? Moreover, HOW "important" features are analysed? A few lines after the reader may understand that the "Gini importance" is involved but this is abruptly introduced and never previously explained in the text. This is relevant since in the discussion there is ample reference to this concept, that should be made clearer to the reader as the ML methods used to highlight predictive power (as stated also in the abstract). Thus, do not limit explanation of the value of "Gini importance" to the methods section but also introduce it in the results when needed.

Having looked closer at the analysis the importance metrics presented are better described as mean decrease in accuracy rather than gini importance and so we have updated the document as such. we have also updated the relevant sections with more detail and,

At line 232 we have added:

To better understand how different features contributed to the predictive power of each classifier we ranked the importance of each feature for each model. Feature importance was calculated by measuring the mean decrease in classifier accuracy when systematically holding out each variable across all tree permutations in a random forest. Feature importance was reported as a mean decrease in accuracy.

At line 532

For each model, feature importance was calculated by measuring the mean decrease in classifier accuracy with the removal of each feature across all trees in the random forest. This metric was reported as the mean decrease in accuracy given the removal of a feature.

Example 2:
In-text lengthy lists of genes in paragraphs 2.6 and 2.7 are confusing and not really useful to support the points which are the link between signature composition and gene mutations, and their enrichment. In my opinion these two paragraphs could be merged into one and gene lists with associated values could be put into tables, leaving commentary and description in the text.

We have extracted the lists of genes and added to a table as suggested at line 282.

Example 3:
The cophenetic correlation obtained on NMF components (lines 198-203) is the metric used to select component signatures: once again it's not explained how these method is able to summarize and group arm ploidy signatures. A simpler yet complete explanation is needed, stating the goal (summarizing the ensemble of arm ploidy signatures into a manageable number to see if kidney cancers are characterized by specific patterns) and the rationale used by the chosen method to do that (NMF, definition of cophenetic score, cluster stability, ...).

We have included an additional paragraph on NMF on line 231.

NMF is a multivariate analysis tool commonly used for easily interpretable decomposition. NMF was chosen because it provides both dimensionality reduction and clustering. Essentially NMF decomposes a feature matrix into two descriptive matrices, a basis, which describes the feature composition of each component and a coefficient, which describes the component composition of each sample in the original feature matrix.

We have also included more detail on cophenentic scores at line 240

Each of these trials provided a cophenentic score, a measure cluster stability, i.e., how well the clusters obtained by NMF preserved the pairwise distances between the original data points.

Other important inconsistencies are:
1) All figures need to be more readable in axes (too small as they are generated)

We have regenerated all image with larger axis text where possible, Please note that the PDF image files attached separately to the manuscript are all vector images and as such they are very clear compared to the embedded versions.

2) Figure 3, The correlation clusters in these heatmaps are difficult to grasp at first sight, due to small characters and lack of the color scale legend

We have increased the size of the axis although we were limited by space. We avoided using a colour scale due to this space constraint, instead we have described the scale in the legend.

3) Supplementary figures 1 and 2 COMPLETELY lack legend. What are these?

We did submit legends with the two supplementary figures:

Supp. figure 1. A ranked bar chart showing the most extreme arm-wise copy number correlation coefficients.

And:

Supp. figure 2. A graphical illustration of arm-wise copy number correlation. Nodes represent chromosome arms and edges represent correlations of above a threshold of r=0.4 and below r=-0.4. Red edges denote positive correlation between the chromosome arms and blue represent negative correlations.

I’ll make sure these are submitted correctly next time and I have attached a copy at the end of the manuscript.

4) ALL S1 tables, as referred to in lines 281, 450, 451, are completely missing. No traces of them either in the main text, or in the supplementary zip archive (or even in the bitbucket repository).

We will make sure to resubmit correctly this file for the supplementary section.

5) Supplementary material in the bitbucket repo is not described at all, just linked-referenced. All supplementary material - if provided - must have a clear description and be linked to main text where is appropriate.

We did not describe this resource very well, instead of supplementary or support material for the paper the bitbucket download link links specifically to the source data required to run the analysis using the previously linked source code. we have updated the text as such.

The source data required for running this analysis, including copy number and mutation data is available at https://bitbucket.org/bioinformatics_lab_sussex/ploidy_nmf/downloads/.

Additional points:
6) In the general narrative it's not clear enough why kidney cancers were chosen as the investigation objects: is there any contradictory data? are these cancers particularly prone to this kind of investigation, and why? In the discussion I found (line 313) an explanation (i.e. distinct and well described karyotype), thus this should immediately made clear in the introduction to justify kidney cancer selection as a manageable proxy for such study.

We have added more justification for choosing to focus on kidney cancers in the introduction:

line 84:

In this study we focused on the analysis of kidney cancers, giving us the opportunity to compare and contrast three molecularly distinct cancers that all arise at the same primary site. In addition, KIRC has a distinct and well characterised karyotype, aiding the validation of our methods.
7) It's not completely clear to me the relevance internal reletionship between of ploidy changes in pairs of chromosomes (par. 2.4). This feature need to be contextualized and biological relevance let emerge (allelic effects? dominant-recessive effects?).

We have added some justification for looking at the correlation of chromosome arm copy number change based on previous observations.

line203

A loss in chromosome arm 3p paired with a gain in chromosome arm 5q is a common and well described trait of KIRC tissue. Next we investigated how the changes of ploidy in other pairs of chromosome arms relate to each other.

Moreover the figure does not help: I guess the two axes (please label them) of each heatmaps refer to two pairs. So this is why the value in the diagonal (bears different colors), differently to what happens when the focus is to the comparisons among arms of different chromosomes. Am I interpreting correctly?

As above we have increased the axis size and we have added labels. we have attempted to describe the reported results along the diagonal in the legend.

8) Figure 4: the two panels (left and right, or a and b) are inverted. Panels should be labelled and put in order coherent with the legend. Moreover the colors used in the two panels are probably from the default ggplot2 palette, but in this case they are internally inconsistent, making the figure a bit ambiguous: in left panel the three colors refer to the kidney cancers, in right panels the same colors do not.

Minor points/typo by line:
- line 47: a leftover underscore "_" fixed.
- lines 155-156 are redundant as a repetition of statements in lines 139-141 Apologies, we haven't been able to find this issue on the lines surrounding 155-156 and 139-141.
- in line 169 "post-cancer", should read "pan-cancer", otherwise the sentence/claim does not make sense to me. Fixed.
- line 205, a leftover "was" at the end of the line Fixed on line 252? “were was”.
- line 217-218: these description could be much clearer if the described object would be pointed in the figure by - for instance - a set of markers or a set of arrows. Apologies, we are not entirely clear which results you suggest we highlight.
- in figure 5 are shown ROCs for performance of random forest: what about the other classifiers cited? Are they consistent?

We did not report scores from the other classifiers as they all performed with much less predictive power compared to Random Forest classifiers. we have included the text:

We trialled four classifiers; Bernoulli naive Bayes, support vector machine, logistic regression and random forest. Due to consistently better performance when compared to the other classifiers all ROC AUC scores below are based on the results of the random forest classifier.

- line 343: typo "kideny" in place of "kidney" fixed.
- line 425: typo "them" in place of "their" fixed.

We have also carefully read over the manuscript and we have made a range of other fixes as highlighted in yellow.

Reviewer 2 Report

The article “Defining signatures of arm-wise copy number change and their associated drivers in kidney cancers” by Graeme Benstead-Hume is an ambitious research article. In this article authors have taken data from TCGA and analyse copy number change patterns in cancer cells. The article is well structured and appropriately written with good statistical analysis.  Some of the minor comments can be included.

Many of the line needs careful reading eg. line no 13, 16, 47 etc KIRC, GDC was not defined in 1st I think line no 31-32 can be rewritten to make statement clear Line no 341 and 342 can be elaborated to highlight the important findings.  

Author Response

Response for Reviewer 2

The article “Defining signatures of arm-wise copy number change and their associated drivers in kidney cancers” by Graeme Benstead-Hume is an ambitious research article. In this article authors have taken data from TCGA and analyse copy number change patterns in cancer cells. The article is well structured and appropriately written with good statistical analysis.  Some of the minor comments can be included.

Many of the line needs careful reading eg. line no 13, 16, 47 etc KIRC, GDC was not defined in 1st I think line no 31-32 can be rewritten to make statement clear Line no 341 and 342 can be elaborated to highlight the important findings.  

We have defined abbreviations correctly

Apologies but my line numbers do not seem to correspond with your comments and so although we have made some changes around these lines we may not have made the specific amendments required.

We have also carefully read over the manuscript and we have made a range of other fixes as highlighted in yellow.

Round 2

Reviewer 1 Report

The Authors have addressed all the issues raised in the first round of revision. The technicalities used have been described in more details and the manuscript is more readable in its current form.

I'm overall satisfied with the editing made by the authors.

Author Response

Thank you for your feedback and for helping us to improve the paper!